# Effect of *Citrus aurantium* Aroma on the Happiness of Pre-Hospital Emergency Staff: A Randomized Controlled Trial

**DOI:** 10.3390/healthcare10122475

**Published:** 2022-12-07

**Authors:** Ali Asghar Ghods, Nemat Sotodeh-asl, Hamid Zia, Raheb Ghorbani, Mohsen Soleimani, Mojtaba Vaismoradi

**Affiliations:** 1Nursing Care Research Center, Semnan University of Medical Sciences, Semnan 3514799422, Iran; 2Department of Psychology, Semnan Branch, Islamic Azad University, Semnan 3514799422, Iran; 3School of Nursing and Midwifery, Semnan University of Medical Sciences, Semnan 3514799422, Iran; 4Social Determinants of Health Research Center, Semnan University of Medical Sciences, Semnan 3514799422, Iran; 5Faculty of Nursing and Health Sciences, Nord University, 8049 Bodø, Norway; 6Faculty of Science and Health, Charles Sturt University, Orange, NSW 2800, Australia

**Keywords:** aromatherapy, *Citrus aurantium*, happiness, healthcare organization, nonpharmacologic method, prehospital emergency care, stress

## Abstract

Happiness is a positive internal experience and an indicator of mental health. Having happy and efficient employees is one of the ideals of healthcare organizations, given its impact on the quality of healthcare services. Emergency healthcare staff members face various unpleasant and stressful events. It has been suggested that fragrant herbs such as *Citrus aurantium* can have cheerfulness effects on individuals. Therefore, this study was conducted to investigate the effect of *Citrus aurantium* aroma on the happiness of pre-hospital emergency staff. A randomized placebo controlled clinical trial was carried out on 167 prehospital emergency medical staff who were randomly assigned into two groups of aromatherapy and placebo. The participants received *Citrus aurantium* aromatherapy and placebo for five work shifts and for two consecutive weeks. Aromatherapy was performed using 10% *Citrus aurantium* scent applied via a pendant containing 1.5 cc of *Citrus aurantium* scent. The distance from the nose to the container was about 20 cm with the neck straight and the head in a balanced position. Data regarding staff happiness were collected using the Oxford Happiness Questionnaire, which were analyzed using descriptive and inferential statistics. The mean level of happiness did not differ between the intervention and control groups (*p* = 0.99). Work experience had a statistically significant inverse association with happiness (coefficient = −0.76, 95% CI: −1.49, −0.03). The findings of this randomized controlled trial indicate that *Citrus aurantium* aroma has no effect on the happiness of prehospital emergency medical services workers. There is a need to study the effect of a combination of complementary and alternative methods on the creation of positive psychological feelings among pre-hospital emergency staff.

## 1. Introduction

Happiness refers to mental happiness, which involves a variety of individuals’ evaluations of oneself and one’s life. These evaluations include life satisfaction, excitement, positive mood, lack of depression, and anxiety [1]. Various aspects of happiness are manifested in people’s cognitions and emotions. Happiness produces energy and enthusiasm, and it can protect individuals against stress, leading to a healthy lifestyle and wellbeing [2,3]. It is an important part of quality of life and is more important than wealth. Plato, the Greek philosopher, described happiness as a state of balance and harmony across three elements of reason, excitement, and desire. Furthermore, Aristotle considered happiness a spiritual life [4]. There is a bilateral relationship between happiness and mental health. Increased happiness leads to increased mental health that consequently enhances happiness [5]. A higher level of happiness increases perceived health among people [6].

A relationship between physical health and mental health based on happiness has been shown [7]. A more efficient immune system and a higher quality of life are the consequences of being happy [8]. Happiness can improve physical health, regardless of how it is acquired. Happy people experience the feeling of security, make decisions more easily, and have a more participatory spirit [9]. Personal, psychological, family, social, and spiritual components are the individual predictors of happiness [10].

Happy people usually evaluate their skills better, remember positive events more than negative ones, and make better decisions in both life and work situations [11]. The ability to handle difficult life situations plays as a protective role against perceived stress and facilitates perceived happiness [12]. The components of life satisfaction, efficiency and sociality, health, positive cognition, feeling of happiness, and self-esteem are important factors influencing happiness [13].

Opportunities to help others contribute to nurses’ happiness at work, but difficult work experiences hinder it [14]. Happiness increases social cohesion and organizational morale, and it enhances employee participation in organizational decisions, leading to improved organizational productivity [15]. The main attribute of job satisfaction is happiness [16], and happy employees are more motivated to do their job [17].

The consequences of happiness among nurses are higher job satisfaction and productivity, more probability of achieving clinical goals, and less job burnout and intention to quit the job [18,19]. Happiness manifesting as internal satisfaction with life enables nurses to craft their tasks [20].

Having happy and efficient employees is one of the ideals of healthcare organizations. Happiness among employees is associated with their productivity, work safety, and job satisfaction [21]. To make nurses happy, ethical support and professional work conditions should be provided by nurse managers [18,22].

Emergency healthcare staff members face various unpleasant and stressful events during their work shifts [23]. Emergency nurses do not often feel happy, mainly due to their tasks and duties [24]. They often observe a patient’s injury, unconsciousness, or even death, which can lead to work pressure and stress [25]. The stress level in emergency healthcare staff is so severe that it can exacerbate their job tension. Emergency healthcare staff often resort to medications such as hypnotic or anxiolytics to control and improve their physical and psychological wellbeing, which can cause dependency and even addiction to medications [26]. Therefore, their job stress should be reduced, and its consequences on their performance should be minimized [27].

Stress reduces individuals’ happiness. Increased levels of stress are associated with decreased happiness, mental health, and quality of life [28,29]. It is believed that stress reduction can provide more opportunities for happiness. It is invigorating and can have booster effects on the brain, lungs, and digestive system [30]. Emergency healthcare staff with a happy mood are more successful in the preservation of their own and their clients’ health [31].

Aromatherapy as a nonpharmacologic method uses an aroma obtained from the extracts of different parts of plants to improve people’s physical and mental wellbeing and mood [32]. Aromatherapy is a gift of nature and a noninvasive method for relieving and controlling physical and psychological symptoms [33]. *Citrus aurantium* is one of the most widely used and native medicinal plants in Iran, which grows in the north and south of Iran. In traditional Iranian medicine, the flowers of this plant are used to treat neurological diseases such as hysteria, seizures, and weakness. It is considered generally safe to use with a potential application to medicinal products in the future [34].

Previous studies have shown that aromatherapy using *Citrus aurantium* can safely reduce anxiety in patients with acute coronary syndrome [35] and relieve fatigue among patients with acute myocardial infarction [36]. This nonaggressive method has been shown to be helpful in the reduction of labor pain [37], improvement of sleep quality among postmenopausal women [38], and relief of emotional symptoms in premenstrual syndrome [39]. Patients undergoing hemodialysis have also experienced a higher quality of life [40]. *Citrus aurantium* is a beneficial medication that can reduce the side-effects of Alzheimer’s disease [41].

Considering the stressful nature of the job of emergency healthcare staff and the need to improve their happiness, this study was conducted to investigate the effect of *Citrus aurantium* on happiness among pre-hospital emergency staff. The research hypothesis was that *Citrus aurantium* could improve happiness among pre-hospital emergency staff.

## 2. Materials and Methods

### 2.1. Design and Participants

This randomized placebo controlled clinical trial was carried out from June 2019 until January 2020. The trial protocol was registered on the Iranian website of clinical trials under the code IRCT20110430006342N9. Moreover, the ethics committee of Semnan University of Medical Sciences (code: IR.SEMUMS.REC.1397.262.167) approved this study and corroborated its ethical considerations.

A convenience sampling method was used to recruit all male pre-hospital emergency staff in urban and intercity roads of Semnan city in Iran that met the inclusion criteria.

Inclusion criteria were having a healthy sense of smell, having at least one year of work experience as emergency staff, no history of asthma and allergies, no unfortunate incident during the month before the intervention, and no use of perfume during the study. Unwillingness to participate in this research and the uncomfortable feeling or sensitivity to the smell of aroma led to exclusion.

The participants were randomly assigned into two groups of *Citrus aurantium* (*n* = 83) and placebo (*n* = 84). The stratified randomization method with a block size of six was used for group assignment based on the samples’ experiences (±5 years) and place of service (urban or intercity roads) (Figure 1). The statistical expert remained blind and generated the random allocation sequence. The main researcher assigned the participants to the groups.

After obtaining the required permissions, the main researcher referred to pre-hospital emergency stations and selected those emergency staff members who met the abovementioned inclusion criteria. They were informed of the research process, their anonymity, and confidentiality of the data, and they were requested to sign a written informed consent form. They were requested not to apply any type of perfume during the study. If any participant used perfume, they were excluded from the intervention on that day. Furthermore, they were asked not to take work shifts on the night before the intervention and take a regular work shift on the day of the intervention.

### 2.2. Intervention

The participants received *Citrus aurantium* or placebo aromatherapy for five work shifts from 8 a.m. to 8 p.m. over a 2 week period. The time period between the interventions was no more than 72 h. Aromatherapy was performed using 10% *Citrus aurantium* essential oil and using a pendant containing 1.5 cc of *Citrus aurantium* essential oil prepared by Barij Essence^®^ Company, Tehran, Iran. The oil was extracted from the flowers using the steam distillation technique. The ingredients of this oil were linalool, linalyl acetate, flavonoids, and tannins. The distance from the container to the nose was about 20 cm with the neck straight and the head in a balanced position for optimum aromatherapy during the participants’ activities.

In the placebo group, the containers were filled with 1.5 cc of distilled water. Inhalation of *Citrus aurantium* did not cause complications or discomfort in the participants. Although the participants were not informed of what was added to the pendants, they could detect the smell of *Citrus aurantium* essential oil. Therefore, blinding of the participants was impossible.

### 2.3. Data Collection

A demographic questionnaire was completed by the participants. It consisted of questions about the participants’ age, height, weight, marriage status, education level, work experience as pre-hospital emergency staff, service location, number of children, and income level.

Data regarding happiness were collected using the Oxford Happiness Questionnaire (OHI), consisting of 29 items with four options. Each item was scored from 0 to 3, with a score range between 0 and 87. A higher score indicates a higher level of happiness.

Furnham et al. reported an alpha coefficient of 90% for the OHI [42]. The reliability coefficient of this tool based on the calculation of Cronbach’s alpha coefficient among nurses working in the emergency department was 0.899 [43]. The validity and reliability of the Farsi version of OHI were confirmed in a previous study with a Cronbach’s α coefficient of 0.92 [44]. Happiness was measured before and after the intervention on the first day, after the last intervention session, and 1 week after the last intervention session.

### 2.4. Data Analysis

Descriptive and inferential statistics were used for data analysis. For the assessment of normality, the Kolmogorov–Smirnov (K–S) test was used. Chi-square, Mann–Whitney U test, and Friedman test were used to compare the groups. The effect size of the intervention was compared between the groups using the calculation of Cohen’s d. Univariate and multivariate linear regression analyses of factors associated with happiness were performed. Included variables in the data analysis model were work experience, number of children, income level, marital status, education level, service location, and group (intervention/placebo). Due to the high collinearity of age and work experience, the age variable was not included in the multivariate linear regression model.

The SPSS software v.21 was used to analyze data, and *p* < 0.05 was considered statistically significant. The statistical experts retained blind to the group assignments to prevent bias during the data analysis. The research was reported according to the Consolidated Standards of Reporting Trials (CONSORT) (see Appendix A).

## 3. Results

There was no attrition in the study. Therefore, data from all recruited participants (*n* = 167) were included in the data analysis. Inhalation of *Citrus aurantium* did not cause any complication or discomfort and the intervention was considered completely safe.

The demographic characteristics of the participants in the groups are shown in Table 1. The mean age ± standard deviation in the intervention group was 29.95 ± 5.43 years, while that in the placebo group was 30.36 ± 5.97 years. The Mann–Whitney U test and chi-square test showed no statistically significant differences in demographic variables between the groups, indicating the groups’ homogeneity (*p* > 0.05).

Happiness before the intervention in the groups had no differences. Happiness in both groups increased significantly over time (*p* < 0.001). However, the difference in happiness between the groups in the data collection times was not statistically significant (Table 2).

Univariate and multivariate linear regression analyses of factors associated with happiness are shown in Table 3. Only work experience had a statistically significant inverse association with happiness (coefficient = −0.76, 95% CI: −1.49, −0.03).

## 4. Discussion

This research aimed to examine the effect of *Citrus aurantium* aroma on the happiness of pre-hospital emergency staff. According to our research findings, the mean scores of happiness did not differ between the intervention and control groups. This indicated that the smell of *Citrus aurantium* had no more effect on the happiness of the pre-hospital emergency staff compared to the placebo group. There has been no research using a similar intervention on emergency staff for comparison with our findings. Therefore, our findings were compared with those of studies on the effect of *Citrus aurantium* on health-related indicators among patients.

Similar to our findings, the study by Karimzadeh (2021) did not support an effect of *Citrus aurantium* on the reduction in pain in conscious intensive care patients [45]. Valehi (2019), in a study on the effect of aromatherapy using *Citrus aurantium* on life expectancy in hemodialysis patients, did not observe any positive effect of aromatherapy [46]. Converse to our findings, Abdollahi et al. (2020), in a study on the effect of aromatherapy using bitter orange extract on sleep quality in patients with type 2 diabetes, showed that aromatherapy improved sleep quality [47]. Zare et al. (2019), also in a study on the effect of *Citrus aurantium* and fluoxetine capsules on postpartum depression in women, concluded that it was more effective in the treatment of postpartum depression compared with the use of fluoxetine capsules and placebo [48].

The contradictory results of the effect of *Citrus aurantium* scent on psychological symptoms can be attributed to the research environment, amount of scent applied, and duration and number of its use. Moreover, it should be noted that pre-hospital emergency staff often report a high level of work stress compared to other people [49]. Therefore, measures used for the reduction of stress among them should be very strong and affective to reduce high levels of work-related pressure and stress.

In our study, work experience had an inverse association with happiness. A cross-sectional study in the USA reported that emergency nurses with more years of experience in practice had a higher level of compassion satisfaction and less burnout [50]. Emergency nurses normally experience stress and work-related exposures that enhance the danger of fatigue and burnout among them [51]. A higher level of confidence in one’s own abilities and more work engagement due to having more job experiences help staff members stay happy [52,53].

Being happy means having no stress. Essential oils have their own sedative properties. Furthermore, smells reminiscent of happy memories can help regenerate happy feelings [54]. *Citrus aurantium* contains compounds such as linanol, linanil acetate, limonene, coumarin, and a variety of flavonoids. Many plant extracts with similar compounds create sedative effects [55].

As a research limitation, standardizing work shift conditions was beyond the research team’s ability. However, efforts were made to exclude work shifts that were associated with a high number of traumatic events for the participants.

## 5. Conclusions

The ineffectiveness of *Citrus aurantium* aroma on the happiness of the pre-hospital emergency staff indicates a need for stronger measures for the creation of good feelings among them under stressful work conditions.

In a situation where pre-hospital emergency staff members face unpleasant work situations, more powerful aroma or a combination of aromas can help with a reduction in their stress and cause happiness. A combination of complementary and alternative methods should be examined to understand their effectiveness in the creation of positive psychological feelings among pre-hospital emergency staff and avoid their dependence on medications.

## Figures and Tables

**Figure 1 healthcare-10-02475-f001:**
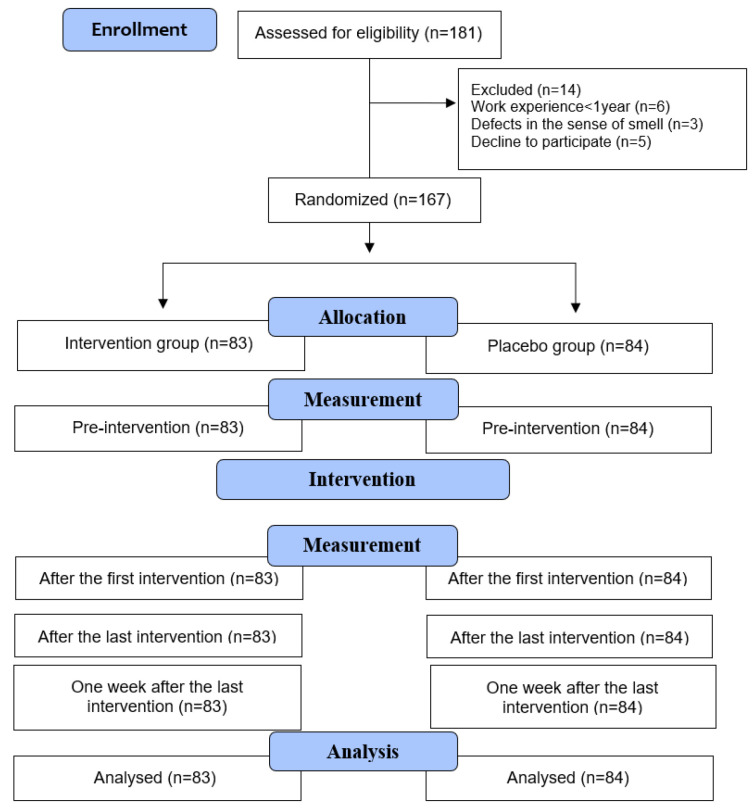
CONSORT flow diagram showing recruitment and progress of the trial.

**Table 1 healthcare-10-02475-t001:** The demographic characteristics of the participants in the groups.

		Aromatherapy		Placebo		
		*n*	%	*n*	%	*p*-Value
Marital status	Married	49	59	46	54.8	0.449
	Single	34	41	37	44	
	Separated	0	0	1	1.2	
Number of children	No child	43	51.8	44	44	0.835
	One	20	24.1	24	28.6	
	Two	17	20.5	14	16.7	
	Three	3	3.6	2	2.4	
	>3	0	0	0	0	
Service location	Intercity road	53	63.9	53	36.9	0.919
	Urban	30	36.1	31	63.1	
Education level	Associate degree	63	75.9	67	79.8	0.446
	Bachelor’s degree	19	22.9	17	20.2	
	Master’s degree and higher	1	1.2	0	0	
Field of education	Medical emergency	65	78.3	64	76.2	0.680
	Nursing	9	10.9	12	14.3	
	Technician of anesthesiology	1	1.2	3	3.6	
	Rescue associate	7	8.4	4	4.8	
	Other	1	1.2	1	1.1	

**Table 2 healthcare-10-02475-t002:** Mean, standard deviation (SD), and median of happiness in the groups.

	Aromatherapy			Placebo			*p*-Value *	Cohen’s d
	Mean	SD	Median (IQR ***)	Mean	SD	Median (IQR ***)		
Happiness before the intervention	45.16	10.50	44 (39, 53)	43.76	10.68	43 (39, 50)	0.425	0.124
Happiness after the first intervention	46.40	9.19	45 (40, 53)	45.44	9.38	45 (40, 50)	0.448	0.118
Happiness after the last intervention	49.90	9.52	50 (44, 55)	48.70	9.01	48 (42, 53)	0.377	0.137
Happiness 1 week after the last intervention	53.71	9.88	51 (48, 60)	53.57	9.58	53 (48, 56)	0.996	0.001
*p*-Value **	<0.001			<0.001				

* Mann–Whitney U test; ** Friedman test; *** interquartile range.

**Table 3 healthcare-10-02475-t003:** Univariate and multivariate linear regression analyses of the study variables.

Characteristics	Univariate		Multivariate	
	Coefficient	95% CI	Coefficient	95% CI
Work experience (year)	−0.58	−0.94, −0.21	−0.76	−1.49, −0.03
Number of children	−1.99	−3.87, −0.11	0.58	−3.52, 4.68
Income level	−1.35	−4.04, 1.34	0.92	−2.29, 4.12
Marital status				
Married	1.45			
Other	1.00	−1.87, 4.77	−0.89	−5.53, 3.75
Education level				
Associate degree	2.93			
Bachelor’s degree or higher	1.00	−1.01, 6.87	0.40	−4.36, 5.15
Service location				
Intercity road	1.00			
Urban	−3.42	−6.80, −0.04	−0.004	−4.68, 4.67
Group				
Intervention	1.63			
Placebo	1.00	−1.65, 4.92	1.65	−1.64, 4.95

CI: confidence interval.

## Data Availability

All data related to this study are included in the article.

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
