# Peer review of "Effect of Citrus aurantium Aroma on the Happiness of Pre-Hospital Emergency Staff: A Randomized Controlled Trial"

_healthcare, 2022, doi:10.3390/healthcare10122475_

Round 1

Reviewer 1 Report

Thanks to the authors for their efforts in conducting the research, but the manuscript has some problems that make it difficult to be accepted.

1. In the introduction section, the authors have limited themselves to giving information about the importance of happiness. While they should deal with the concept of happiness and its components and scope and its impact on the job.

2. The research done in this field and their results should be added in the introduction.

3. The cited references are not relevant to some statements. For example: “Ancient physicians have recommended Citrus aurantium to increase vitality and happiness [25].

4. In method section the sample size calculation, process of randomization and concealment should be addressed.

5. The questionnaire used for outcome measure (OHI) is in English. The process of translation and face and content validity of questionnaire should be explained by authors.

6. Which software was used for statistical analysis?

7. Discussion needs to be expanded.  The discussion section is one of the final parts of a research paper, in which the authors should describes, analyzes, and interprets their findings. They should explain the significance of those results and tie everything back to the research question(s).

Author Response

Dear Editor,

Thank you for the provision of this opportunity to revise and resubmit the article. We are also thankful for the comments by the reviewers that helped us to improve the quality of our article. Here they are the details of changes that have been highlighted using red color in the text.

Sincerely/Authors

Reviewer 1

Comments and Suggestions for Authors

Thanks to the authors for their efforts in conducting the research, but the manuscript has some problems that make it difficult to be accepted.

  1. In the introduction section, the authors have limited themselves to giving information about the importance of happiness. While they should deal with the concept of happiness and its components and scope and its impact on the job.

Answer: More details on happiness and its impact on job were added.

  1. The research done in this field and their results should be added in the introduction.

Answer: There is no previous research on the use of Citrus aurantium aroma on the happiness of pre-hospital emergency staff. We have introduced some studies on the effect of Citrus aurantium aroma om patients instead.

  1. The cited references are not relevant to some statements. For example: “Ancient physicians have recommended Citrus aurantium to increase vitality and happiness [25].

Answer: The citation was deleted and others were also checked.

  1. In method section the sample size calculation, process of randomization and concealment should be addressed.

Answer: The related details were added. There was no sampling formula as the convenience method was used for recruitment, but they were assigned randomly to the groups.

  1. The questionnaire used for outcome measure (OHI) is in English. The process of translation and face and content validity of questionnaire should be explained by authors.

Answer: The instrument in the Farsi version was available and its psychometric properties have been assessed in a previous study.

  1. Which software was used for statistical analysis?

Answer: The SPSS software v.21 was used to analyze data.

  1. Discussion needs to be expanded.  The discussion section is one of the final parts of a research paper, in which the authors should describes, analyzes, and interprets their findings. They should explain the significance of those results and tie everything back to the research question(s).

Answer: We have tried to compare our findings. Given the lack of similar studies, there was no similar research to be used for this aim.

Reviewer 2 Report

Authors found out ineffectiveness of aroma thepary on the happiness of pre-hospital emegrency staff working in urben area in Iran. I suggest to publish the paper.

  1. The paper discusses effect of citrus aurantium on happiness of pre- hospital emergency staff
  2. and I consider that their topic is relevant to their research area
  3. it shows that the environmental and other factors such as stress and their work environment can affect their happiness at work
  4.   the study design is well prepared with good amount of participants 
  5. authors discussed and cited relevant papers for their justification

Author Response

Dear Editor,

Thank you for the provision of this opportunity to revise and resubmit the article. We are also thankful for the comments by the reviewers that helped us to improve the quality of our article. Here they are the details of changes that have been highlighted using red color in the text.

Sincerely/Authors

Reviewer 2

Comments and Suggestions for Authors

Authors found out ineffectiveness of aroma thepary on the happiness of pre-hospital emegrency staff working in urben area in Iran. I suggest to publish the paper.

  1. The paper discusses effect of citrus aurantium on happiness of pre- hospital emergency staff
  2. and I consider that their topic is relevant to their research area
  3. it shows that the environmental and other factors such as stress and their work environment can affect their happiness at work
  4.   the study design is well prepared with good amount of participants 
  5. authors discussed and cited relevant papers for their justification

Answer: Thanks for your support to our article. The introduction section has been improved. 

Reviewer 3 Report

The oil was placed 20 cm below the nose as a verum. In the case of paramedics working outdoors and with a high degree of movement, it should also be mentioned in the discussion to what extent the aroma reaches the nose at all during this activity.

Author Response

Dear Editor,

Thank you for the provision of this opportunity to revise and resubmit the article. We are also thankful for the comments by the reviewers that helped us to improve the quality of our article. Here they are the details of changes that have been highlighted using red color in the text.

Sincerely/Authors

Reviewer 3

Comments and Suggestions for Authors

The oil was placed 20 cm below the nose as a verum. In the case of paramedics working outdoors and with a high degree of movement, it should also be mentioned in the discussion to what extent the aroma reaches the nose at all during this activity.

Answer: The scent from Essential oils is strong to be smelled from a distance when staff do their job activities. Therefore, they could easily smell the scent and notice its presence when they breath. This is the common routine of aromatherapy found in various related textbooks.

Round 2

Reviewer 1 Report

.

Author Response

Thanks for your support to out article.